# Recycling of Lead Pastes from Spent Lead–Acid Batteries: Thermodynamic Constraints for Desulphurization

**Yongliang Xiong**

Nuclear Waste Disposal Research & Analysis, Sandia National Laboratories (SNL), 1515 Eubank Boulevard SE, Albuquerque, NM 87123, USA; yxiong@sandia.gov

**Abstract:** Lead–acid batteries are important to modern society because of their wide usage and low cost. The primary source for production of new lead–acid batteries is from recycling spent lead–acid batteries. In spent lead–acid batteries, lead is primarily present as lead pastes. In lead pastes, the dominant component is lead sulfate ($PbSO_4$, mineral name anglesite) and lead oxide sulfate ($PbO \bullet PbSO_4$, mineral name lanarkite), which accounts for more than 60% of lead pastes. In the recycling process for lead–acid batteries, the desulphurization of lead sulfate is the key part to the overall process. In this work, the thermodynamic constraints for desulphurization via the hydrometallurgical route for recycling lead pastes are presented. The thermodynamic constraints are established according to the thermodynamic model that is applicable and important to recycling of lead pastes via hydrometallurgical routes in high ionic strength solutions that are expected to be in industrial processes. The thermodynamic database is based on the Pitzer equations for calculations of activity coefficients of aqueous species. The desulphurization of lead sulfates represented by $PbSO_4$ can be achieved through the following routes. (1) conversion to lead oxalate in oxalate-bearing solutions; (2) conversion to lead monoxide in alkaline solutions; and (3) conversion to lead carbonate in carbonate solutions. Among the above three routes, the conversion to lead oxalate is environmentally friendly and has a strong thermodynamic driving force. Oxalate-bearing solutions such as oxalic acid and potassium oxalate solutions will provide high activities of oxalate that are many orders of magnitude higher than those required for conversion of anglesite or lanarkite to lead oxalate, in accordance with the thermodynamic model established for the oxalate system. An additional advantage of the oxalate conversion route is that no additional reductant is needed to reduce lead dioxide to lead oxide or lead sulfate, as there is a strong thermodynamic force to convert lead dioxide directly to lead oxalate. As lanarkite is an important sulfate-bearing phase in lead pastes, this study evaluates the solubility constant for lanarkite regarding the following reaction, based on the solubility data, $PbO \bullet PbSO_4 + 2H^+ \rightleftharpoons 2Pb^{2+} + SO_4^{2-} + H_2O(l)$.

**Keywords:** lead–acid batteries; desulphurization; thermodynamic model

## 1. Introduction

Lead has been used for more than ~2500 years [1] and today lead is still very important to various modern industries. Among them, lead–acid batteries have been proven to be essential in various applications in electric vehicles, energy storage, lighting and ignition batteries, and uninterrupted power supplies, etc. Battery production accounts for more 80% of lead consumption [2]. In production of lead– acid batteries, about 70–80% of the lead needed for battery manufacture comes from recycling of spent lead–acid batteries, and the rest of the lead is produced from processing the lead ores from mining. Therefore, the majority of the lead used for battery production is sourced from spent lead–acid batteries.

In spent lead–acid batteries, the major components include plastic containers, lead alloy grids, waste acids, and pastes. Among them, lead pastes and lead alloy grids are recycled for lead. In lead pastes, the mineralogical compositions in an order of decreasing weight percentage based on semi-quantitative analyses include anglesite

($PbSO_4$, 38%), lanarkite ($PbO \bullet PbSO_4$, 36%), plattnerite ($PbO_2$, 9%), lead oxide sulfate hydrate [$(PbO)_3(Pb(SO_4))(H_2O)$, 6%], leadhillite [$Pb_4(SO_4)(CO_3)_2(OH)$, 4%], scrutiny ($PbO_2$, 3%], litharge (PbO, 2%), and lead (Pb, 2%) [3]. Lanarkite was also observed by other researchers [4]. The chemical components include lead sulfates ($PbSO_4$ and $PbO \bullet PbSO_4$) (57.21 to 60.57 wt.%), lead dioxide ($PbO_2$) (5.66 to 26.70 wt.%), lead monoxide (PbO) (13.08 to 29.53 wt.%), and metallic lead (Pb) (1.24 to 4.5 wt.%) [5,6]. In spent lead–acid batteries, lead sulfates are primarily produced via the following reactions,

$$Pb + SO_4{}^{2-} \rightleftharpoons PbSO_4(cr) + 2e^- \tag{1}$$

$$PbO_2 + 4H^+ + SO_4{}^{2-} + 2e^- \rightleftharpoons PbSO_4(cr) + 2H_2O(l) \tag{2}$$

$$PbO_2 + Pb + 2H_2SO_4 \rightleftharpoons 2PbSO_4(cr) + 2H_2O(l) \tag{3}$$

$$PbO + PbSO_4 \rightleftharpoons PbO \bullet PbSO_4 \tag{4}$$

During the operation of a lead–acid battery, Reaction (1) in the forward direction primarily occurs in a discharge period, and metallic lead behaves as an anode. Reaction (2) in the forward direction takes place during discharge and lead dioxide acts as a cathode. In a recharging period, Reactions (1) and (2) operate in the reverse direction.

In recycling lead pastes, the presence of lead sulfates presents environmental, health hazard, and economic problems. This is because the decomposition of lead sulfates is energy-intensive, as it requires high temperatures exceeding 1000 °C [7,8], and the decomposition process using coal or coke as the fuel produces many gaseous and solid pollutants, such as lead fume as well as dilute $SO_2$ gas streams [7–9]. Therefore, in order to prevent the above environmental and health problems as well as the energy-intensive problem, lead sulfates must be desulphurized first [10,11]. In the hydrometallurgical recycling process for lead–acid batteries, there are three desulphurization processes of lead pastes with oxalate, carbonate, and alkaline solutions. The desulphurized lead products (i.e., lead oxalate, lead hydroxide, and lead carbonate) are then smelted to produce lead ingots.

In this work, the thermodynamic constraints for desulphurization of lead paste via three routes in oxalate, carbonate, and alkaline solutions are presented. The thermodynamic constraints are constructed primarily based on the thermodynamic models established previously [12–15], in addition to the thermodynamic parameters from the literature. These thermodynamic models use the Pitzer equations for calculations of activity coefficients for aqueous species and are valid to high ionic strengths. Therefore, these thermodynamic models are suitable and ideal for such applications. Based on the optimum condition for each desulphurization route and their products according to the thermodynamic constraints, more efficient and eco-friendly recycling processes for lead–acid batteries could be examined.

## 2. Thermodynamic Models for Lead Species

Xiong et al. [12] established the thermodynamic models for the interactions of lead with oxalate to high ionic strengths, based on the experimental studies on solubilities of lead oxalate [$PbC_2O_4(cr)$] in various media, including NaCl, $K_2C_2O_4$, and the mixtures of $KNO_3 + K_2C_2O_4$. In that work, both Pitzer interaction parameters and the Specific Ion Interaction Theory (SIT) parameters for lead species with oxalate are determined. Subsequently, a thermodynamic model for the interactions of lead species in carbonate solutions was developed [14]. After that, a thermodynamic model for lead species in alkaline solutions was developed [15]. It should be emphasized that the above-mentioned models utilize some parameters including Pitzer parameters from the literature, in addition to those determined in the above-mentioned studies. Those parameters from the literature are detailed and cited in Table 1.

In the following thermodynamic analyses, the computer code EQ3/6 Version 8.0a [16,17] with the thermodynamic database DATA0.FM2 [18–20] was employed for the thermodynamic calculations. The software was originally created by Thomas J. Wolery at Evanston,

Illinois, USA. The database contains those parameters related to lead species, as tabulated in Tables 1 and 2.

**Table 1.** Equilibrium constants at infinite dilution at 25 °C and 1 bar for key chemical species relevant to recycling of lead pastes from lead–acid batteries.

| Reactions | log $K_s^o$, and Cumulative Formation Constants, log $\beta_1^o$, log $\beta_2^o$, log $\beta_3^o$ | Reference and Remarks |
|---|---|---|
| $PbCO_3(cr) \rightleftharpoons Pb^{2+} + CO_3^{2-}$ | $-13.76 \pm 0.15\ (2\sigma)$ | [14] |
| $Pb^{2+} + CO_3^{2-} \rightleftharpoons PbCO_3(aq)$ | $6.87 \pm 0.09\ (2\sigma)$ | [21] |
| $Pb^{2+} + 2CO_3^{2-} \rightleftharpoons Pb(CO_3)_2^{2-}$ | $10.41 \pm 0.18\ (2\sigma)$ | [22] |
| $Pb^{2+} + CO_3^{2-} + Cl^- \rightleftharpoons Pb(CO_3)Cl^-$ | $7.23 \pm 0.74\ (2\sigma)$ | [21] |
| $PbO(cr) + 2H^+ \rightleftharpoons Pb^{2+} + H_2O(l)$ | 12.59 | [23], EQ3/6 database |
| $Pb^{2+} + H_2O(l) \rightleftharpoons PbOH^+ + H^+$ | $-7.46$ | [24] |
| $Pb^{2+} + 2H_2O(l) \rightleftharpoons Pb(OH)_2(aq) + 2H^+$ | $-17.05 \pm 0.10\ (2\sigma)$ | [15] |
| $Pb^{2+} + 3H_2O(l) \rightleftharpoons Pb(OH)_3^- + 3H^+$ | $-27.99 \pm 0.15\ (2\sigma)$ | [15] |
| $PbC_2O_4(cr) \rightleftharpoons Pb^{2+} + C_2O_4^{2-}$ | $-11.13 \pm 0.15\ (2\sigma)$ | [12] A |
| $Pb^{2+} + C_2O_4^{2-} \rightleftharpoons PbC_2O_4(aq)$ | $5.85 \pm 0.10\ (2\sigma)$ | [12] A |
| $Pb^{2+} + 2C_2O_4^{2-} \rightleftharpoons Pb(C_2O_4)_2^{2-}$ | $8.05 \pm 0.15\ (2\sigma)$ | [12] A |
| $Pb^{2+} + C_6H_5O_7^{3-} \rightleftharpoons PbC_6H_5O_7^-$ | 7.28 | [19] |
| $Pb^{2+} + Cl^- \rightleftharpoons PbCl^+$ | 1.48 | [25] |
| $Pb^{2+} + 2Cl^- \rightleftharpoons PbCl_2(aq)$ | 2.03 | [25] |
| $Pb^{2+} + 3Cl^- \rightleftharpoons PbCl_3^-$ | 1.86 | [25] |
| $PbSO_4(cr) \rightleftharpoons Pb^{2+} + SO_4^{2-}$ | $-7.78$ | [26] |
| $PbO \bullet PbSO_4(cr) + 2H^+ \rightleftharpoons Pb^{2+} + SO_4^{2-} + H_2O(l)$ | $2.66 \pm 0.05$ | This study |
| $Na_2C_2O_4(cr) \rightleftharpoons 2Na^+ + C_2O_4^{2-}$ | $-2.61 \pm 0.05$ | This study |
| $K_2C_2O_4(cr) \rightleftharpoons 2K^+ + C_2O_4^{2-}$ | $-1.00 \pm 0.06$ | This study |
| $H_2C_2O_4\ (aq) \rightleftharpoons 2H^+ + C_2O_4^{2-}$ | $-6.07$ | [13] |
| $HC_2O_4^- \rightleftharpoons H^+ + C_2O_4^{2-}$ | $-4.36$ | [13] |
| $PbO_2(cr) + 2H^+ \rightleftharpoons Pb^{2+} + H_2O(l) + 0.5O_2(g)$ | 7.75 | [27] |

A Notice that in Xiong et al. [12], two sets of equilibrium constants were presented. One set is consistent with the Pitzer model, whereas the other set is consistent with the Specific Ion Interaction Theory (SIT) model, and the equilibrium constants from these two sets are different. The equilibrium constants presented in this work are from the set that is consistent with the Pitzer model.

**Table 2.** Pitzer interaction parameters at 25 °C and 1 bar for key chemical species relevant to recycling of lead pastes from lead–acid batteries via hydrometallurgical routes.

| Pitzer Binary Interaction Parameters | | | | | |
|---|---|---|---|---|---|
| Species $i$ | Species $j$ | $\beta^{(0)}$ | $\beta^{(1)}$ | $C^\varphi$ | Reference |
| $Na^+$ | $Pb(CO_3)_2^{2-}$ | 0.4168 | 1.74 | $-0.3161$ | [14] |
| $Na^+$ | $Pb(CO_3)Cl^-$ | 0.2419 | 0.29 | $-0.1802$ | [14] |
| $Na^+$ | $Pb(OH)_3^-$ | 0.3354 | 0.29 | 0 | [15] |
| $Na^+$ | $PbCl_3^-$ | $-0.0605$ | 0 | 0.091 | [12] |
| $Pb^{2+}$ | $Cl^-$ | 0.26 | 1.64 | 0.088 | [25] |
| $PbCl^+$ | $Cl^-$ | 0.15 | 0 | 0 | [25] |
| $K^+$ | $Pb(C_2O_4)_2^{2-}$ | 0 | $-1.86 \pm 0.20$ | $0.198 \pm 0.09$ | [12] |
| $Na^+$ | $Pb(C_2O_4)_2^{2-}$ | 0 | $-1.86 \pm 0.20$ | $0.198 \pm 0.09$ | [12] |
| $PbNO_3^+$ | $NO_3^-$ | $-0.75$ | 0.34 | 0 | [12] |
| $Na^+$ | $C_2O_4^{2-}$ | $-0.2770$ | 1.74 | 0.122 | [13] |
| $K^+$ | $C_2O_4^{2-}$ | $-0.2770$ | 1.74 | 0.122 | This work, using $Na^+/C_2O_4^{2-}$ from [13] as an analog |
| $Na^+$ | $PbC_6H_5O_7^-$ | 0.535 | 0.29 | 0.0196 | [20] |
| $Mg^{2+}$ | $PbC_6H_5O_7^-$ | 1.97 | 1.74 | 0.0771 | [20] |
| $Na^+$ | $C_6H_5O_7^{3-}$ | 0.0877 | 5.22 | 0.047 | [28] |
| $Mg^{2+}$ | $C_6H_5O_7^{3-}$ | 0.9330 | 4.4 | 0 | [29] |

| Pitzer Mixing Interaction Parameters and Interaction Parameters Involving Neutral Species | | | | | |
|---|---|---|---|---|---|
| Species $i$ | Species $j$ | Species $k$ | $\lambda_{ij}$ or $\theta_{ij}$ | $\zeta_{ijk}$ | Reference |
| $HCO_3^-$ | $Pb(CO_3)_2^{2-}$ | | 0.2956 | | [14] |
| $CO_3^{2-}$ | $Pb(CO_3)_2^{2-}$ | | 0.2707 | | [14] |
| $Cl^-$ | $PbCO_3(aq)$ | | $-0.02$ | | [21] |
| $Na^+$ | $PbCO_3(aq)$ | $Cl^-$ | 0 | $-0.145$ | [21] |
| $SO_4^{2-}$ | $Pb(OH)_2(aq)$ | | $-0.5581$ | | [15] |
| $SO_4^{2-}$ | $Pb(OH)_3^-$ | | $-0.4046$ | | [15] |
| $Na^+$ | $Pb^{2+}$ | | 0.10 | | [30] |
| $Cl^-$ | $PbCl_2(aq)$ | | $-0.14 \pm 0.04$ | | [12] |
| $Na^+$ | $PbCl_2(aq)$ | | $-0.11$ | | [30] |

## 3. Results

The thermodynamic analyses for desulphurization of lead pastes primarily focus on three routes: (a) desulphurization in oxalate-bearing media to convert lead sulfates into lead oxalate; (b) desulphurization in carbonate solutions to convert lead sulfates into lead

carbonates; and (c) desulphurization in alkaline solutions to convert lead sulfates into lead oxide. The thermodynamic analytical results for the above three routes are depicted separately in the following.

### 3.1. Desulphurization in Oxalate-Bearing Media

Previous studies have demonstrated from the aspect of thermodynamics and field observations that lead oxalate is a stable phase in the presence of oxalate [12,31,32]. Therefore, desulphurization of lead pastes using oxalate media is a viable route [33,34]. In the presence of oxalate, the major components of lead pastes are subject to the following reactions,

$$PbSO_4(cr) + C_2O_4{}^{2-} \rightleftharpoons PbC_2O_4(cr) + SO_4{}^{2-} \tag{5}$$

$$PbO\bullet PbSO_4(cr) + 2H^+ + 2C_2O_4{}^{2-} \rightleftharpoons 2PbC_2O_4(cr) + SO_4{}^{2-} + H_2O(l) \tag{6}$$

$$PbO(cr) + 2H^+ + C_2O_4{}^{2-} \rightleftharpoons PbC_2O_4(cr) + H_2O(l) \tag{7}$$

$$PbO_2(cr) + 2H^+ + C_2O_4{}^{2-} \rightleftharpoons PbC_2O_4(cr) + H_2O(l) + 0.5O_2(g) \tag{8}$$

$$4PbO_2(cr) + PbS + 8H^+ + 4SO_4{}^{2-} \rightleftharpoons 5PbSO_4(cr) + 4H_2O(l) \tag{9}$$

Reaction (5) represents the conversion of lead sulfate to lead oxalate. The conversion of lead oxide sulfate (lanarkite) to lead oxalate is outlined by Reaction (6). As lead pastes also contain lead monoxide and lead dioxide, Reactions (7) and (8) illustrate the possible reactions for lead monoxide and lead dioxide in the presence of oxalate. Notice that no additional reductant is needed or used in Reaction (8). As detailed below, Reaction (8) is thermodynamically strongly favored. Alternatively, Reaction (9) depicts the use of lead sulfide (PbS) as a reductant to convert lead dioxide to lead sulfate first, which is then converted to lead oxalate, as demonstrated by Reaction (5). Glucose is also suggested as a reductant for reduction of lead dioxide [35].

The solubility constant for PbO•PbSO$_4$(cr) (lanarkite) is not available in the literature. Therefore, the solubility constant for lanarkite in the following form,

$$PbO\bullet PbSO_4(cr) + 2H^+ \rightleftharpoons Pb^{2+} + SO_4{}^{2-} + H_2O(l) \tag{10}$$

is evaluated from the solubility data from [36], using the EQ3/6 Version 8.0a [16,17] with the database DATA0.FM2 [18–20]. The solubility constant is tabulated in Table 1.

According to Reaction (5), the stability fields of PbC$_2$O$_4$(cr) and PbSO$_4$(cr) in the space of activities of sulfate in logarithmic units versus activities of oxalate in logarithmic units (i.e., $\log a_{SO_4^{2-}}$ versus $\log a_{C_2O_4^{2-}}$) are constructed in Figure 1. Figure 1 shows that there is a strong thermodynamic driving force for the conversion of PbSO$_4$(cr) to PbC$_2$O$_4$(cr). For instance, even at a high sulfate activity of 10, an oxalate activity of $10^{-2}$ or higher will convert lead sulfate into lead oxalate.

Figure 2 shows the stability fields of lead oxalate and lead oxide sulfate in the space of pH versus activity of oxalate in logarithmic units (i.e., pH versus $\log a_{C_2O_4^{2-}}$) at various activities of sulfate. The figure suggests that the conversion of lead oxide sulfate to lead oxalate at sulfate activity ranging from 0.001 to 1 with water activity of 1 is thermodynamically strongly favored, especially in acidic pH range, requiring activities of oxalate as low as $10^{-11}$.

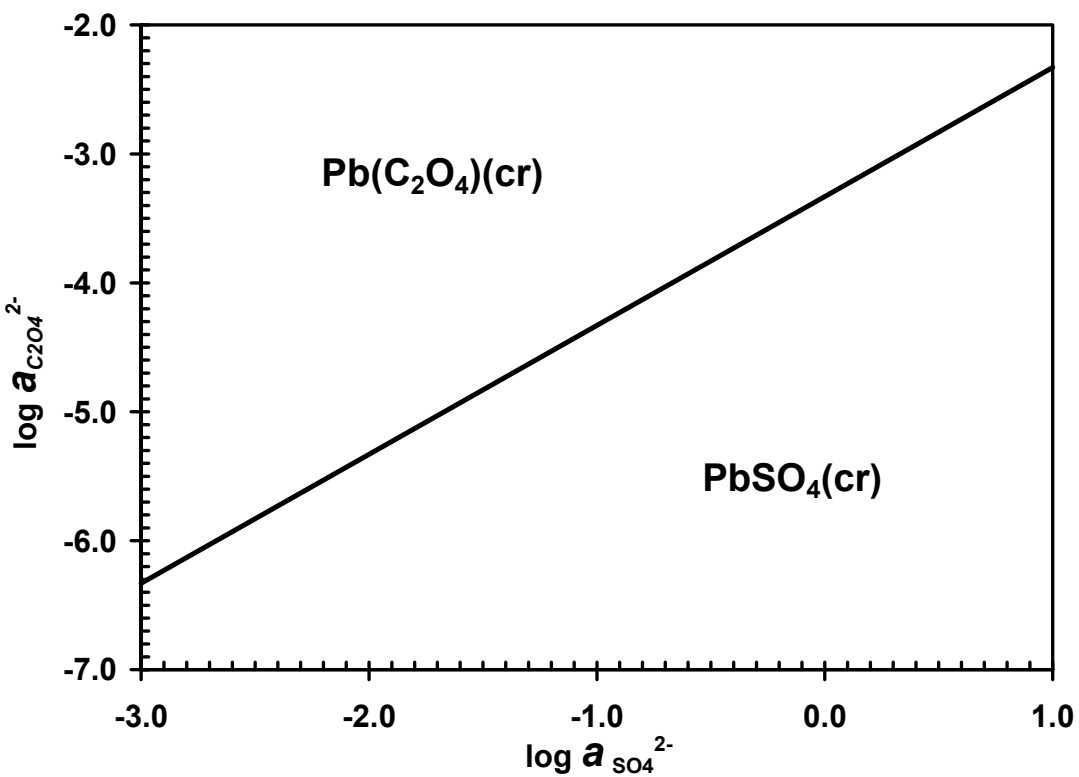

**Figure 1.** The stability fields of lead sulfate [PbSO$_4$(cr), anglesite] and lead oxalate [PbC$_2$O$_4$(cr)] at 25 °C in the space of log $a_{SO_4^{2-}}$ versus log $a_{C_2O_4^{2-}}$ in an oxalate-bearing solution for desulphurization of lead pastes.

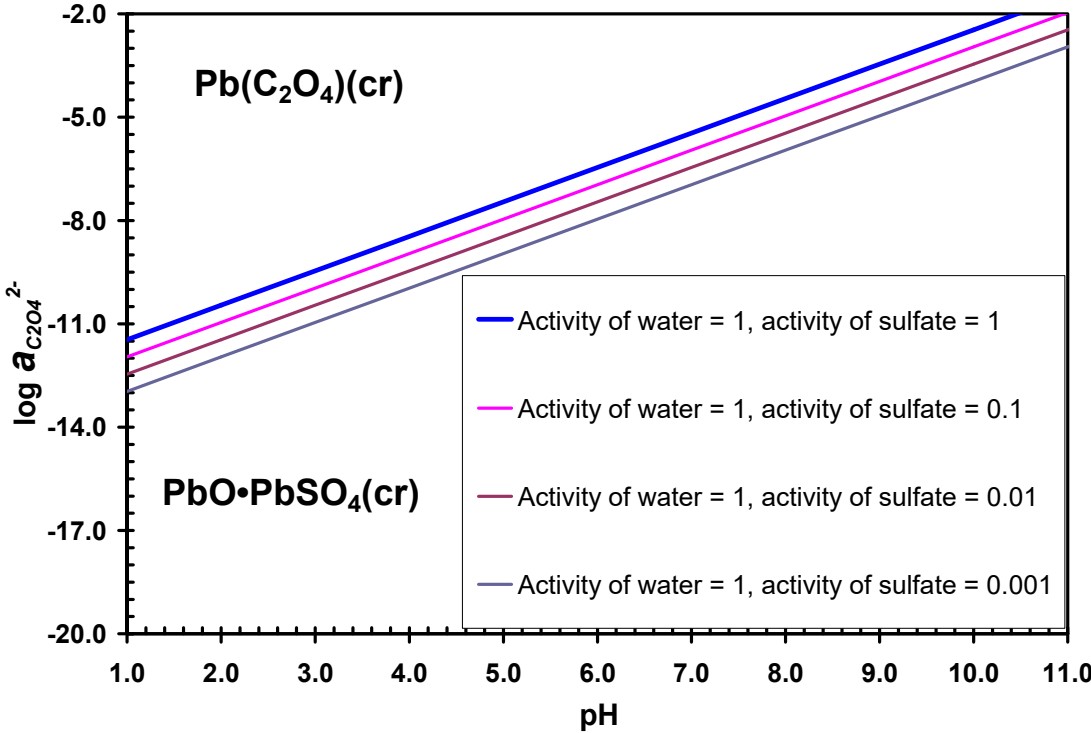

**Figure 2.** The stability fields of lead oxide sulfate [PbO•PbSO$_4$(cr), lanarkite] and lead oxalate [PbC$_2$O$_4$(cr)] at 25 °C in the space of pH versus log $a_{C_2O_4^{2-}}$ in an oxalate-bearing solution for desulphurization of lead pastes at various sulfate (SO$_4^{2-}$) activities ranging from 0.001 to 1 and at water activity equal to 1.0.

As lead monoxide is also present in lead pastes, it will react with oxalate when oxalate-bearing solutions are used for desulphurization, illustrated by Reaction (7). Figure 3 shows the stability fields of lead oxalate and lead monoxide in the space of pH versus activity of oxalate in logarithmic units (i.e., pH versus $\log a_{C_2O_4^{2-}}$) at water activities of 1.0, 0.75, and 0.45. Figure 3 suggests that the stability fields at these three water activities do not deviate significantly. Lead monoxide usually buffers solutions in alkaline pH range [15]. In alkaline pH range, solutions with activity of oxalate higher than $10^{-2}$ will convert lead monoxide into lead oxalate (Figure 3).

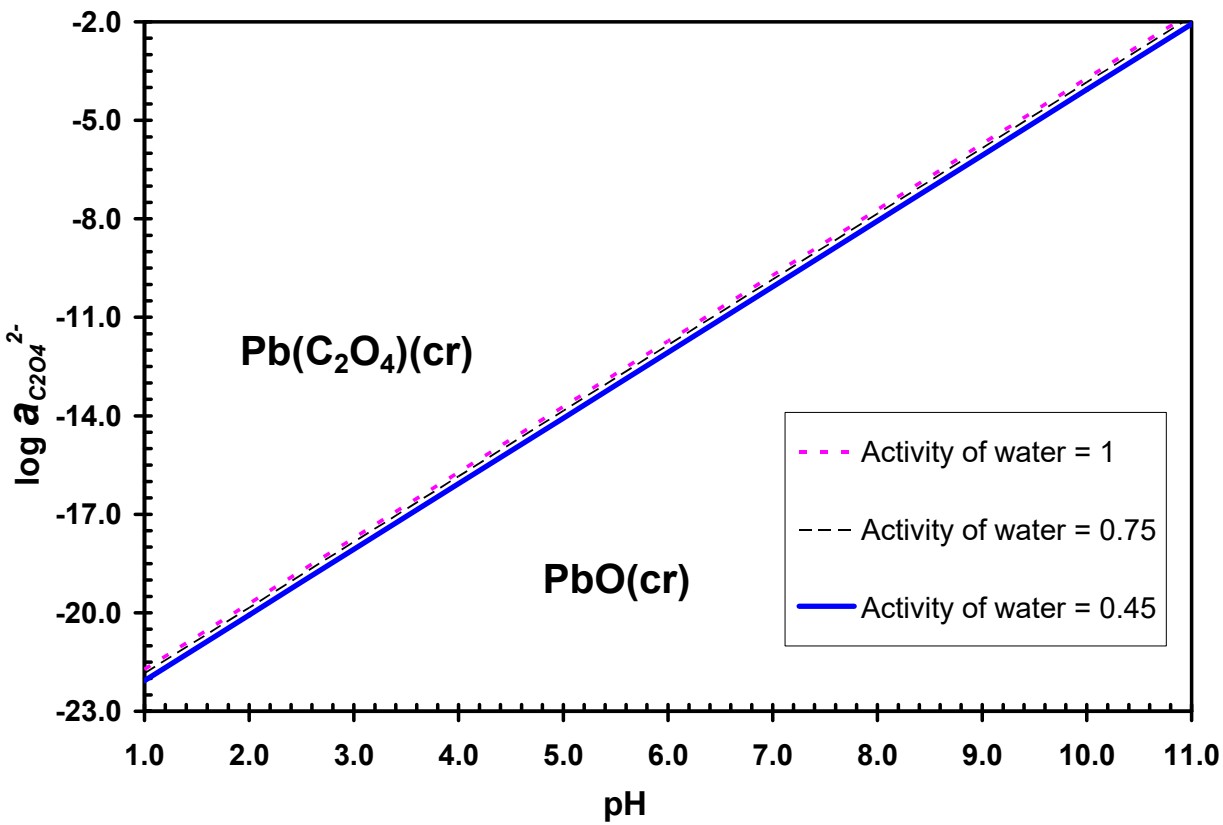

**Figure 3.** The stability fields of lead monoxide [PbO(cr)] and lead oxalate [PbC$_2$O$_4$(cr)] at 25 °C in the space of pH versus $\log a_{C_2O_4^{2-}}$ in an oxalate-bearing solution for desulphurization of lead pastes at water activities of 1.0, 0.75, and 0.45.

Lead dioxide (plattnerite and scrutiny) is present in lead pastes. Usually, reductants are required to reduce it first to lead monoxide. Figure 4 illustrates the direct conversion of lead dioxide into lead oxalate without an added reductant in the space of pH versus activity of oxalate in logarithmic units (i.e., pH versus $\log a_{C_2O_4^{2-}}$) at water activity of 1.0 and various fugacities of oxygen. In Figure 4, two oxygen fugacities are assumed: $f_{O_2} = 0.21$ bars and $f_{O_2} = 10$ bars. In the first scenario, the reduction process is open to the atmosphere. In the second scenario, the reduction process happens in a closed system, such as in a pressure vessel. In the second scenario, the partial pressures of oxygen are allowed to be accumulated up to 10 bars. Figure 4 suggests that lead dioxide can be reduced in oxalate-bearing solutions, as the required activities of oxalate are very low. For instance, the required activities of oxalate in acidic pH 1–3 ranges are less than $10^{-11}$. As demonstrated by Figure 5, the activities of oxalate (i.e., $a_{C_2O_4^{2-}}$) in oxalic acid solutions (~$10^{-4}$) are many orders of magnitude higher than the required activities of oxalate.

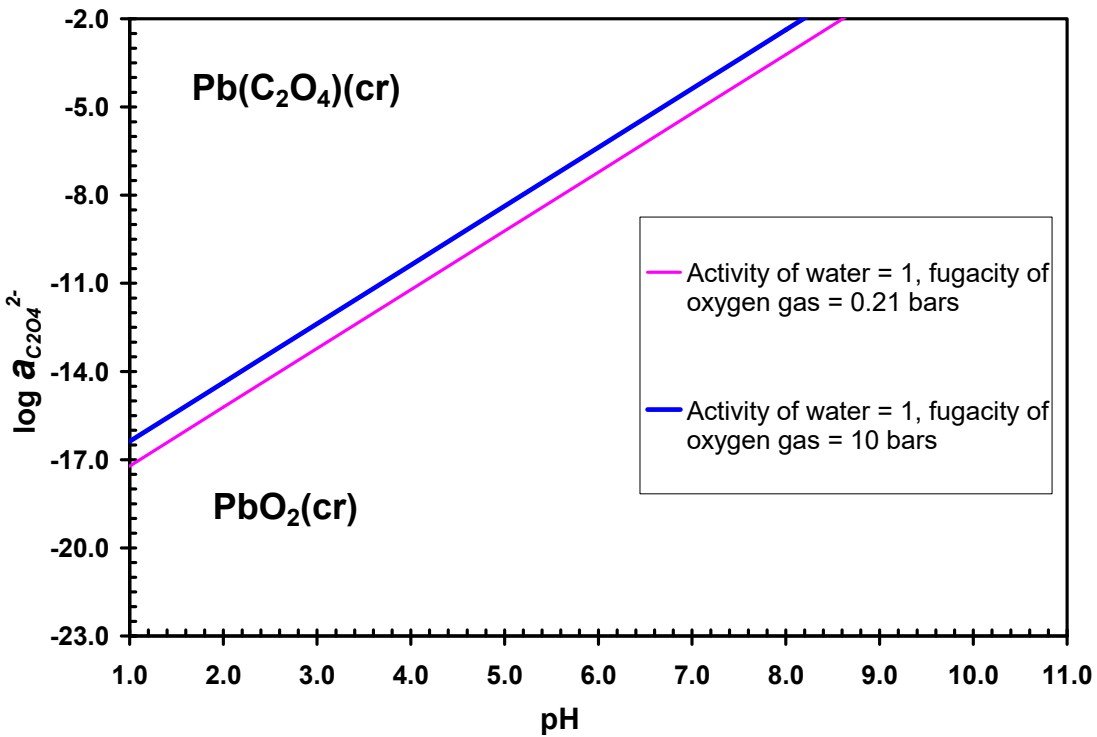

**Figure 4.** The stability fields of lead dioxide [PbO$_2$(cr)] and lead oxalate [PbC$_2$O$_4$(cr)] at 25 °C in the space of pH versus log $a_{C_2O_4^{2-}}$ in an oxalate-dominated solution for desulphurization of lead pastes at water activity of 1.0 with two oxygen fugacities ($f_{O_2}$) at 0.21 bars and 10 bars, respectively.

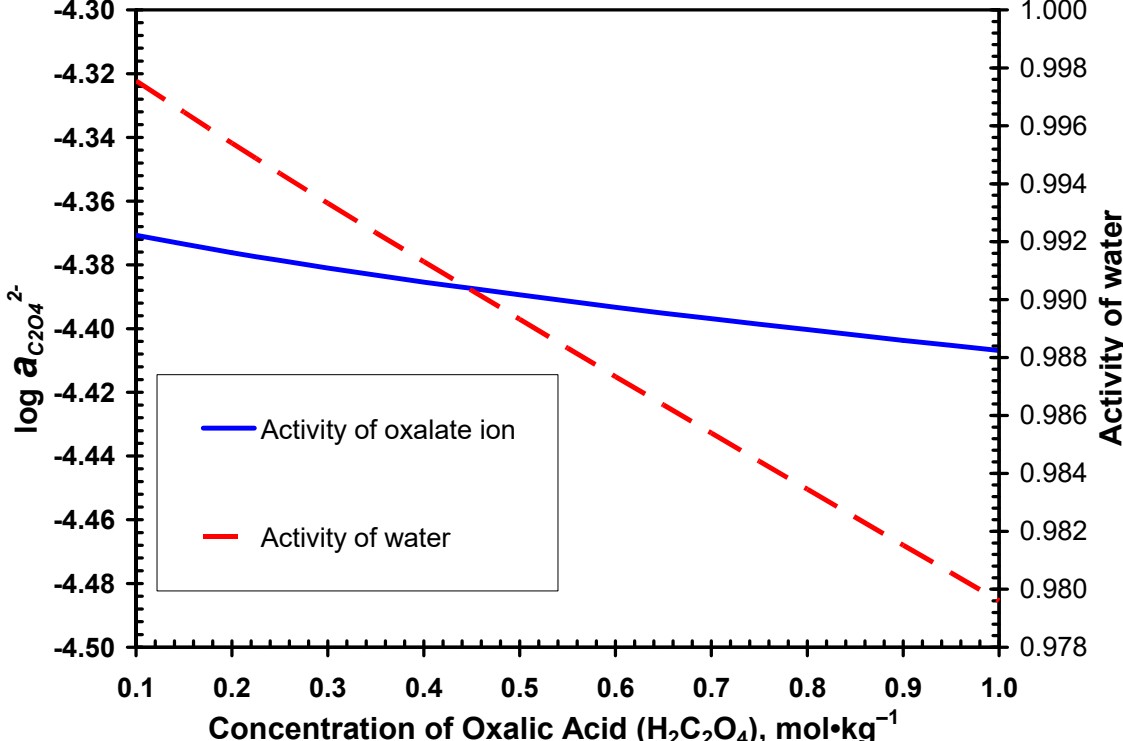

**Figure 5.** Predicted activities of oxalate species and water for oxalic acid (H$_2$C$_2$O$_4$) solutions relevant to desulphurization of lead sulfate in oxalate-bearing solutions at 25 °C and pH$_m$ = 1.0 (negative logarithm of hydrogen ion concentration on molal scale). Activities of oxalate species and water activities of oxalic acid solutions are calculated according to the thermodynamic parameters from Thakur et al. (2015), which are listed in Tables 1 and 2.

Lead sulfide (PbS) has been suggested as a candidate reductant for reduction of lead dioxide [3]. Figure 6 presents the stability fields of the assemblage of lead dioxide and lead sulfide with respect to lead sulfate in the space of pH versus activity of sulfate (i.e., pH versus $\log a_{SO_4^{2-}}$) at water activities of 1.0, 0.75, and 0.45. As suggested by Figure 5, the reduction of lead dioxide to lead sulfate in the presence of lead sulfide is thermodynamically strongly favored, especially in acidic pH range. The reduction process requires low activities of sulfate, less than $10^{-25}$, even at alkaline pH range.

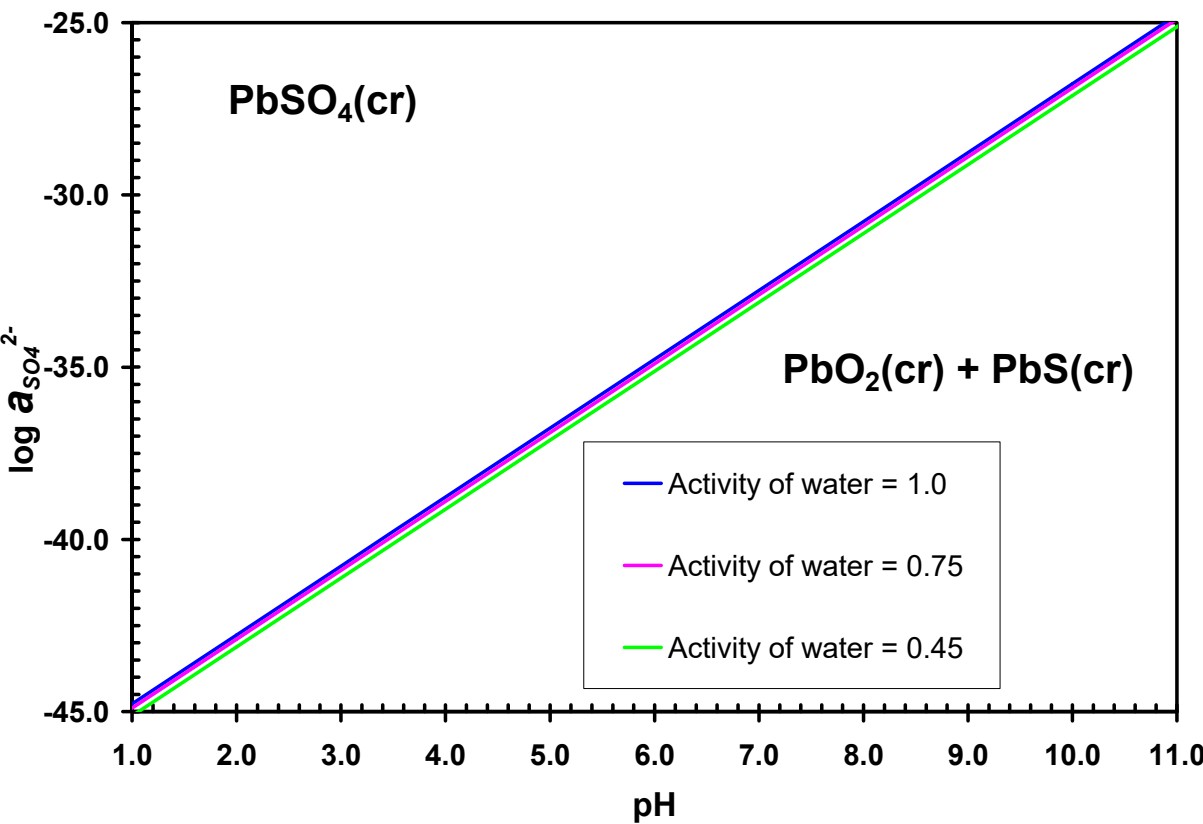

**Figure 6.** The stability fields of the assemblage of lead dioxide [PbO$_2$(cr)] + lead sulfide [PbS(cr)] with respect to lead sulfate [PbSO$_4$(cr)] at 25 °C in the space of pH versus $\log a_{SO_4^{2-}}$ in a sulfate-bearing solution for reduction of lead dioxide in lead pastes at water activities of 1.0, 0.75, and 0.45, respectively.

However, the reduction of lead dioxide to lead sulfate using lead sulfide as a reductant produces more lead sulfates that require desulphurization. In comparison, the use of oxalic acid solution to convert lead dioxide directly to lead oxalate is much simpler, and it does not need to desulphurize additional amounts of lead sulfate, and it is a preferred route. However, the efficacy of the route with oxalic acid solutions depends on the kinetics. If the kinetics are not favored at 25 °C, the temperature can be increased to enhance the kinetics.

As depicted in Figures 1–4, activities of oxalate are the key thermodynamic driving force for the desulphurization process. The oxalate-bearing solutions that are suitable for desulphurization include oxalic acid, potassium oxalate and sodium oxalate solutions. In Figure 5, activities of oxalate species and water activities for oxalic acid solutions up to the saturation point with respect to H$_2$C$_2$O$_4$•2H$_2$O(cr) are predicted based on the thermodynamic model in [13] for the oxalate system. The up-limits of oxalate activities that can be provided by oxalic acid solutions are $10^{-4.4}$ in the acidic pH range. As shown in Figures 1–4, activities of oxalate required for conversion of lead sulfate and lead oxide sulfate in acidic pH range are many orders of magnitude lower than $10^{-4.4}$.

Other oxalate-bearing solutions, including Na$_2$C$_2$O$_4$ and K$_2$C$_2$O$_4$ solutions, can also be used for desulphurization. Based on the experimental work from [37], the saturated concentrations of Na$_2$C$_2$O$_4$ and K$_2$C$_2$O$_4$ in water at 25 °C are 0.27 mol•kg$^{-1}$ and 2.06 mol•kg$^{-1}$,

respectively. According to their solubility data, the solubility constants for $Na_2C_2O_4$ and $K_2C_2O_4$ are determined according to the following reactions,

$$Na_2C_2O_4(cr) \rightleftharpoons 2Na^+ + C_2O_4{}^{2-} \tag{11}$$

$$K_2C_2O_4(cr) \rightleftharpoons 2K^+ + C_2O_4{}^{2-} \tag{12}$$

The solubility constants for $Na_2C_2O_4$ and $K_2C_2O_4$ are tabulated in Table 1. These solubility constants are evaluated in consistency with the thermodynamic model for the oxalate system from [13].

The activities of potassium ion, oxalate species, and water for $K_2C_2O_4$ solutions are computed using the thermodynamic model of [13] for the oxalate system (Figure 7). The concentrations of $K_2C_2O_4$ solutions are up to the saturation point with $K_2C_2O_4(cr)$ (Figure 7). The activities of oxalate in $K_2C_2O_4$ solutions are much higher than those in oxalic acid. Notice that activities of oxalate for $Na_2C_2O_4$ solutions are not separately computed. The first reason for this is that $Na_2C_2O_4(cr)$ has much lower solubility (i.e., 0.27 mol•kg$^{-1}$), which is lower than that of $K_2C_2O_4(cr)$ by about a factor of 8 (2.06 mol•kg$^{-1}$). Second, the interaction parameters for $K^+$—$C_2O_4{}^{2-}$ are assumed to be the same as those for $Na^+$—$C_2O_4{}^{2-}$ (see Table 1). Therefore, the activities of oxalate computed for $K_2C_2O_4$ solutions can be used for $Na_2C_2O_4$ solutions.

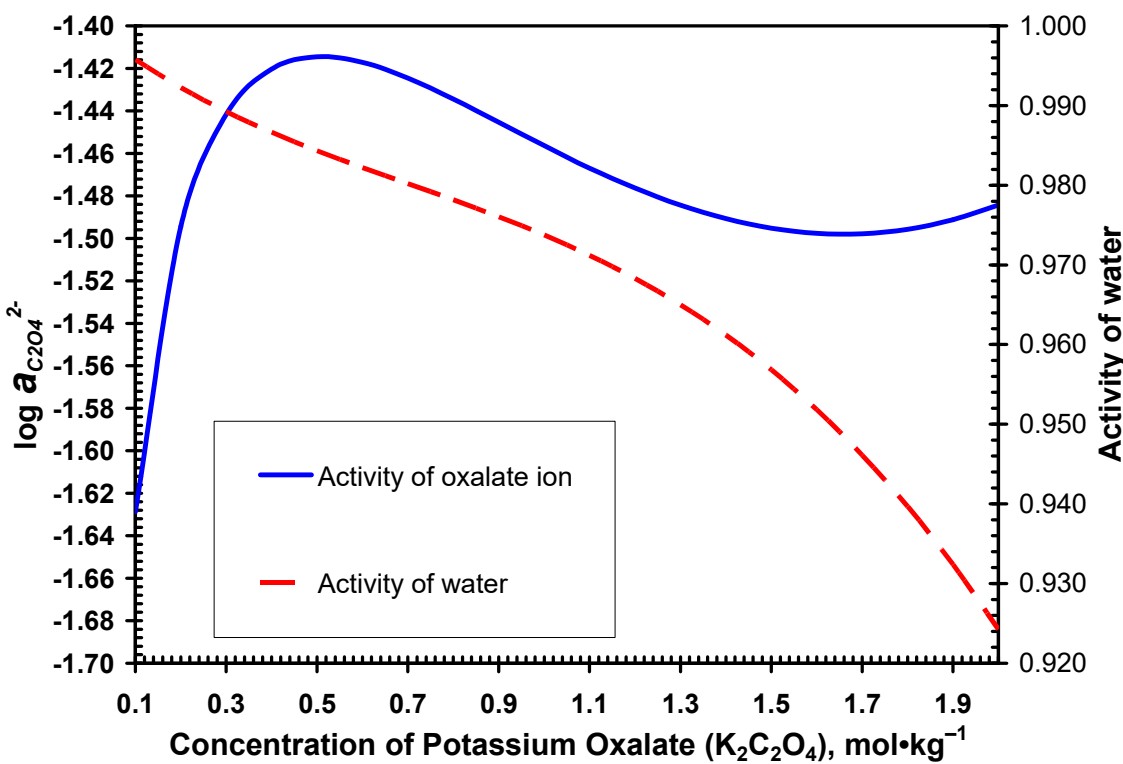

**Figure 7.** Predicted activities of oxalate species and water for potassium oxalate ($K_2C_2O_4$) solutions relevant to desulphurization of lead sulfate in oxalate-bearing solutions at 25 °C and pH$_m$ = 6.5 (negative logarithm of hydrogen ion concentration on molal scale). Activities of oxalate species and water activities of potassium oxalate solutions are calculated according to the thermodynamic parameters from [13]), which are listed in Tables 1 and 2.

### 3.2. Desulphurization in Alkaline Solutions

Alkaline solutions, such as NaOH, have been suggested to be agents for desulphurization [6], and exhibit a higher desulphurization rate and efficiency than that

using carbonate media, such as $Na_2CO_3$ [38]. In alkaline solutions, lead sulfate is desulphurized, as the hydroxyl ion displaces the sulfate ion,

$$PbSO_4(cr) + 2OH^- \rightleftharpoons PbO(cr) + H_2O(l) + SO_4^{2-} \tag{13}$$

Figure 8 shows the stability fields of lead monoxide and lead sulfate in the space of activity of sulfate in logarithmic units versus activity of the hydroxyl ion in logarithmic units (i.e., $\log a_{SO_4^{2-}}$ versus $\log a_{OH^-}$). Figure 8 suggests that lead sulfate will be converted to lead monoxide at various activities of the hydroxyl ion. In low activities of sulfate (i.e., $\leq 10^{-3}$), an activity of the hydroxyl ion at $10^{-5}$ or higher is sufficient to desulphurize lead sulfate by converting it into lead monoxide.

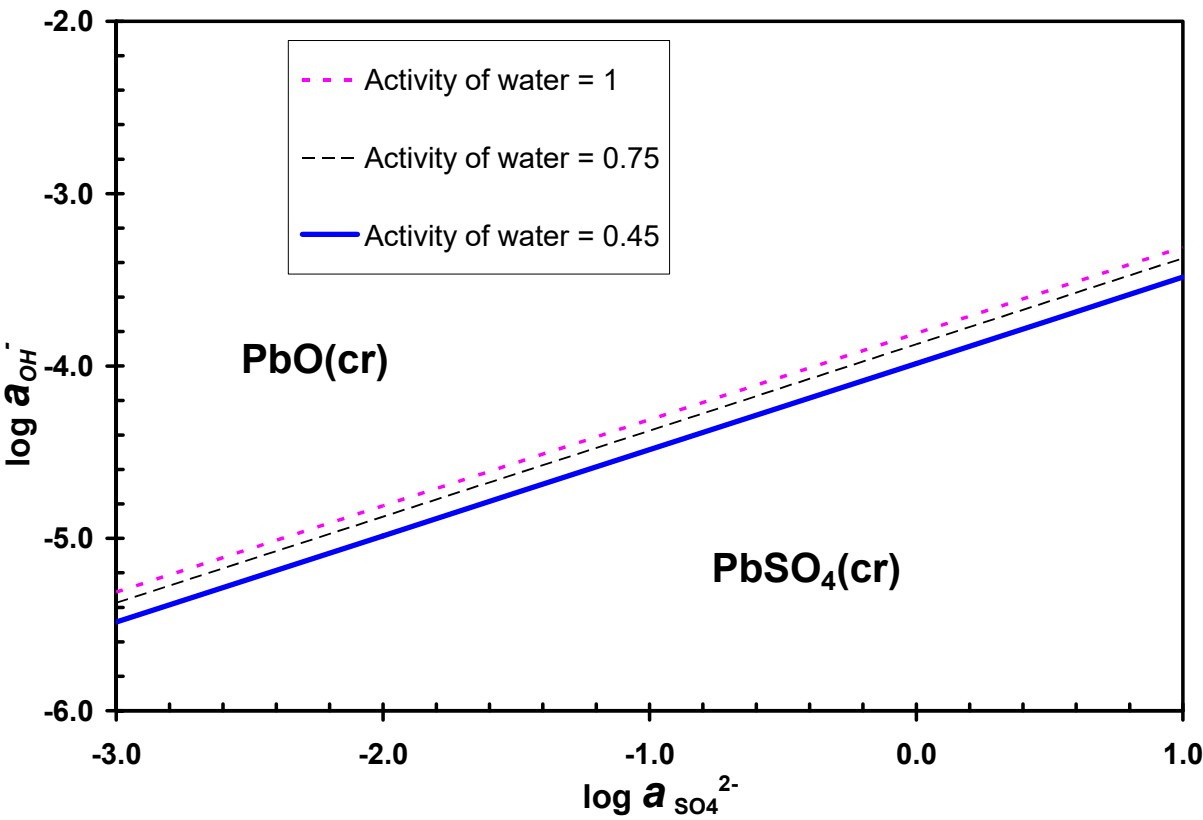

**Figure 8.** The stability fields of lead dioxide [PbO(cr)] and lead sulfate [PbSO$_4$(cr)] at 25 °C in the space of $\log a_{SO_4^{2-}}$ versus $\log a_{OH^-}$ in alkaline solutions for desulphurization of lead pastes at water activities of 1.0, 0.75, and 0.45, respectively.

The desulphurization of lead oxide sulfate in alkaline solutions can be expressed as,

$$PbO\bullet PbSO_4(cr) + 2OH^- \rightleftharpoons 2PbO(cr) + H_2O(l) + SO_4^{2-} \tag{14}$$

Figure 9 displays the stability fields of lead monoxide and lead oxide sulfate (lanarkite) in the space of activity of sulfate in logarithmic units versus activity of the hydroxyl ion in logarithmic units (i.e., $\log a_{SO_4^{2-}}$ versus $\log a_{OH^-}$). In comparison with Figure 6, Figure 9 suggests that the desulphurization of lanarkite requires higher activities of the hydroxyl ion than that required for the desulphurization of lead sulfate at the same activities of sulfate. For instance, at $a_{SO_4^{2-}} = 10^{-3}$, the minimum activity of the hydroxyl ion should be $10^{-4.4}$ for the desulphurization of lanarkite (Figure 9), which is about one order of magnitude higher than that required for the desulphurization of lead sulfate (Figure 8).

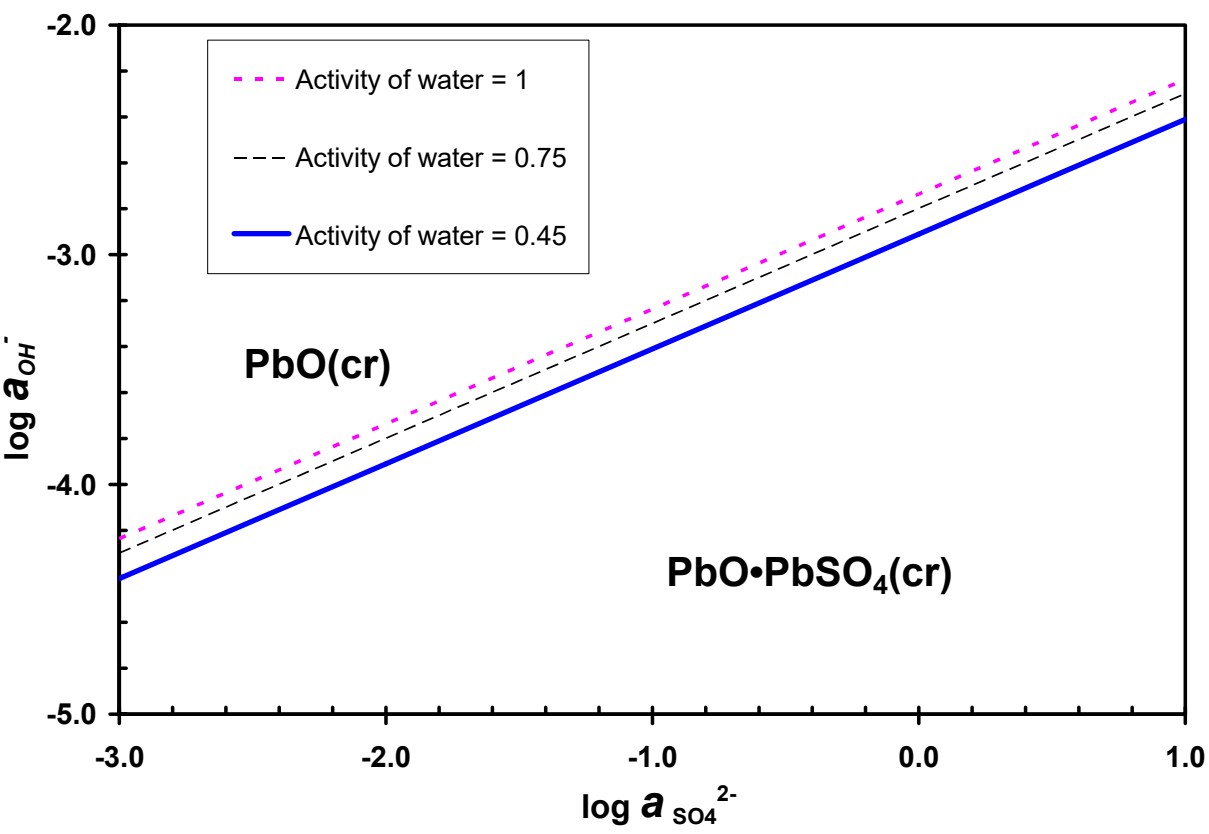

**Figure 9.** The stability fields of lead dioxide [PbO(cr)] and lead oxide sulfate [PbO•PbSO$_4$(cr)] at 25 °C in the space of $\log a_{SO_4^{2-}}$ versus $\log a_{OH^-}$ in alkaline solutions for desulphurization of lead pastes at water activities of 1.0, 0.75, and 0.45, respectively.

In the desulphurization process using alkaline solutions such as NaOH and KOH solutions, the other function of alkaline solutions is to leach lead from lead pastes by dissolution of the lead solid phase. In the above thermodynamic calculations as summarized by Figures 8 and 9 as well as the experimental work in [15], the solubility-controlling phase for lead in alkaline solutions is lead monoxide (PbO, litharge).

Xiong et al. [15] have developed a high precision thermodynamic model for the solubility of litharge in alkaline solutions. Figure 10 represents solubilities of litharge in alkaline solutions in comparison with the experimental solubility data of litharge from Randall and Spencer [39], and Garrett et al. [40] in NaOH and KOH solutions at 25 °C. Notice that the experimental data presented in Figure 10 were not used in the model development in [15]. Figure 10 demonstrates that the model of Xiong et al. [15] can accurately predict the solubility of litharge in alkaline solutions with OH$^-$ concentrations up to 0.25 mol•kg$^{-1}$. Therefore, the model of Xiong et al. [15] can be used for predictions of solubility of lead monoxide in the desulphurization process in alkaline solutions up to OH$^-$ = 0.25 mol•kg$^{-1}$.

One issue with alkaline solutions for desulphurization may be that a reductant must be used to reduce lead dioxide to lead monoxide. This is because the conversion of lead dioxide to lead monoxide is not thermodynamically favored without a reductant,

$$PbO_2(cr) \rightleftharpoons PbO(cr) + 0.5O_2(g) \tag{15}$$

The Gibbs free energy change ($\Delta_r G^0_{298.15}$) for the above reaction is 28.40 kJ•mol$^{-1}$, calculated from the thermodynamic properties from [27] for the respective species in Reaction (11). The positive Gibbs free energy change for Reaction (15) suggests that there is no thermodynamic driving force for lead dioxide to be decomposed into lead monoxide.

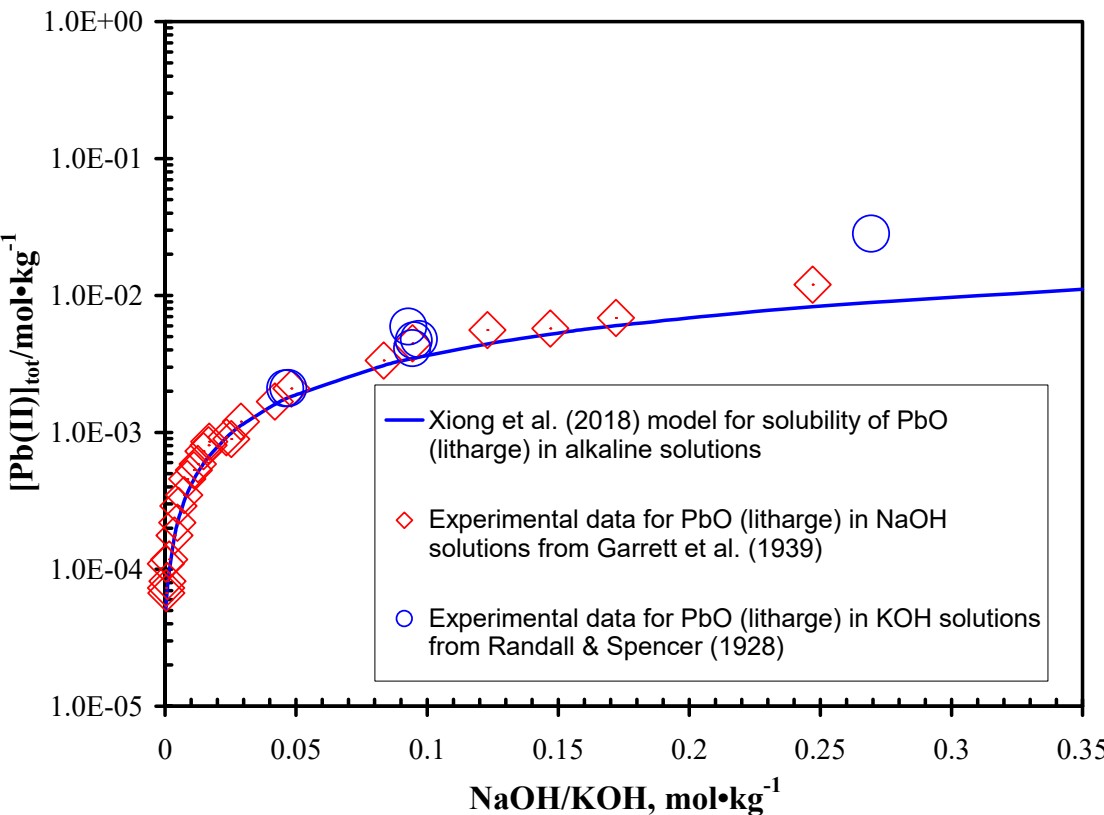

**Figure 10.** Solubilities of lead monoxide (PbO, litharge) as a function of hydroxyl concentrations in dilute and moderate concentration ranges at 25 °C, applicable to desulphurization and leaching of lead pastes with alkaline solutions. The experimental data are from Randall and Spencer [39] and Garrett et al. [40]. The calculated solubilities are based on the model of Xiong et al. [15].

In addition to the above-mentioned reductants (e.g., PbS), an environmentally friendly reductant, glucose ($C_6H_{12}O_6$), has also recently proposed [41]. In their work, lead pastes were first reduced with glucose at 175 °C. After the reduction process, only PbO•PbSO$_4$ and PbSO$_4$ were present. Then, the lead pastes containing only PbO•PbSO$_4$ and PbSO$_4$ were desulphurized in NaOH solutions [41].

In order to facilitate the applications, the solution chemical parameters including activities of water, the hydroxyl ion, and the hydrogen ion for NaOH solutions with concentrations ranging from very dilute (e.g., 0.0001 mol•kg$^{-1}$) to 0.26 mol•kg$^{-1}$ are presented in Figure 11. The activities of the hydroxyl ion in very dilute NaOH solutions are higher than $10^{-4.0}$, which are sufficient for desulphurization of both PbSO$_4$ and PbO•PbSO$_4$ at low sulfate activity of $10^{-3.0}$ (see Figures 8 and 9). However, as the industrial process may use alkaline solutions with higher hydroxyl concentrations, the extension of the lead solubility model of Xiong et al. (2018) to higher hydroxyl concentrations is in progress.

Notice that in the model of Xiong et al. (2018) for alkaline solutions, the speciation schema for lead hydroxyl species includes Pb$^{2+}$, PbOH$^+$, Pb(OH)$_2$(aq), and Pb(OH)$_3^-$. The tetra-hydroxyl lead species, Pb(OH)$_4^{2-}$, determined by Perera et al. (2001), is expected to be important in hyperalkaline solutions. When the model of Xiong et al. (2018) is extended to hyperalkaline solutions, the tetra-hydroxyl lead species will be included.

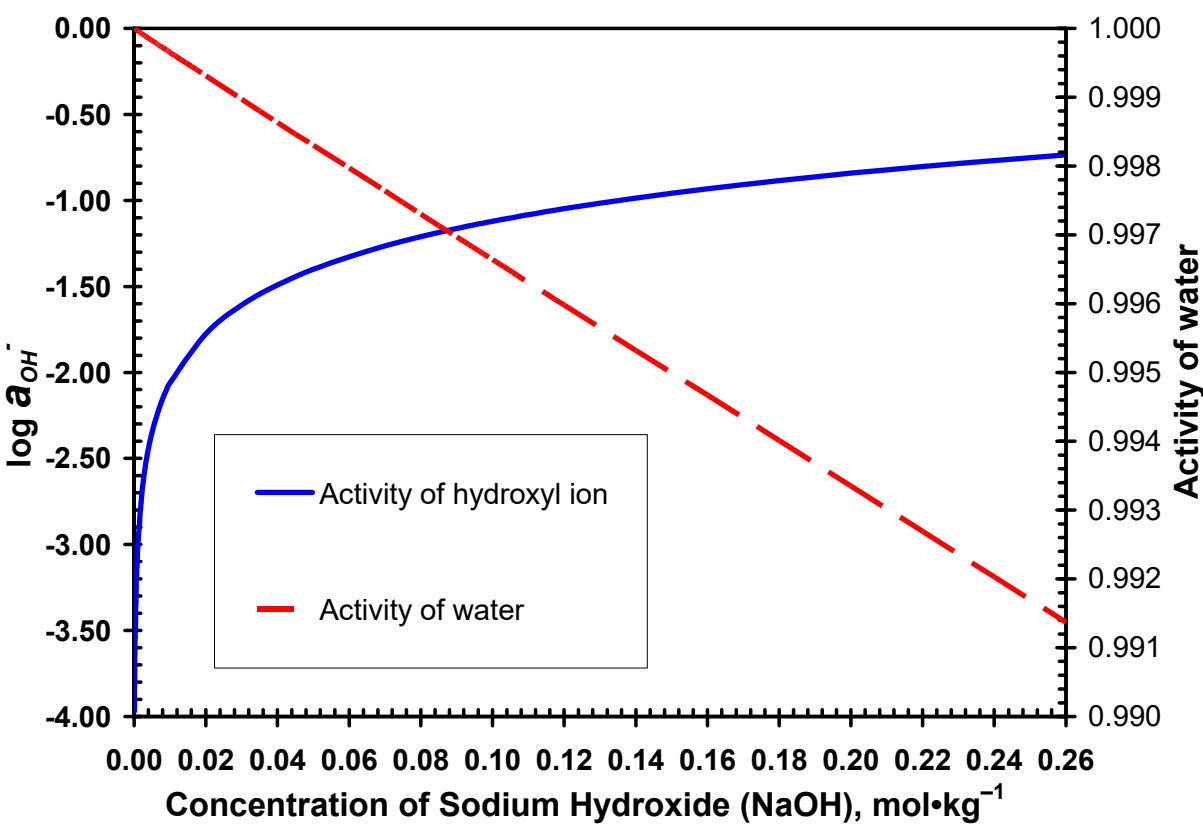

**Figure 11.** Predicted activities of the hydroxyl ion and water for sodium hydroxide (NaOH) solutions relevant to desulphurization of lead sulfate in alkaline solutions at 25 °C. Activities of the hydroxyl ion and water of sodium hydroxide solutions are calculated according to the thermodynamic parameters from Xiong et al. [15].

### 3.3. Desulphurization in Carbonate Media

Cerussite ($PbCO_3$) is the dominant lead phase in carbonate-bearing solutions [14,42,43]. Therefore, lead pastes can be desulphurized in carbonate-bearing solutions by taking advantage of the stability of cerussite in such solutions [44]. The replacement of lead sulfate by cerussite can be cast as the following reaction

$$PbSO_4(cr) + CO_3{}^{2-} \rightleftharpoons PbCO_3(cr) + SO_4{}^{2-} \tag{16}$$

Figure 9 displays the stability fields of cerussite and lead sulfate in the space of activity of sulfate in logarithmic units versus activity of carbonate in logarithmic units (i.e., $\log a_{SO_4^{2-}}$ versus $\log a_{CO_3^{2-}}$). Even in an environment with high activities of sulfate (e.g., $a_{SO_4^{2-}} = 10$), the required activities of carbonate are low (e.g., $10^{-5}$) (Figure 12).

The conversion of lanarkite to cerussite can be expressed as,

$$PbO\bullet PbSO_4(cr) + 2H^+ + 2CO_3{}^{2-} \rightleftharpoons 2PbCO_3(cr) + SO_4{}^{2-} + H_2O(l) \tag{17}$$

Figure 13 illustrates the stability fields of lanarkite and cerussite in the space of activity of sulfate in logarithmic units versus activity of carbonate in logarithmic units (i.e., $\log a_{SO_4^{2-}}$ versus $\log a_{CO_3^{2-}}$). The stability relations are calculated at an alkaline pH 10 and water activity of 1.0. Figure 13 shows that the desulphurization of lanarkite into cerussite requires higher activities of carbonate at the same activities of sulfate, in comparison with the desulphurization of anglesite (c.f. Figures 12 and 13). For instance, at $a_{SO_4^{2-}} = 10^{-3}$, the desulphurization of lanarkite into cerussite requires a carbonate activity of $\geq 10^{-6.6}$, in com-

parison with a carbonate activity of $\geq 10^{-9}$ for desulphurization of anglesite into cerussite.

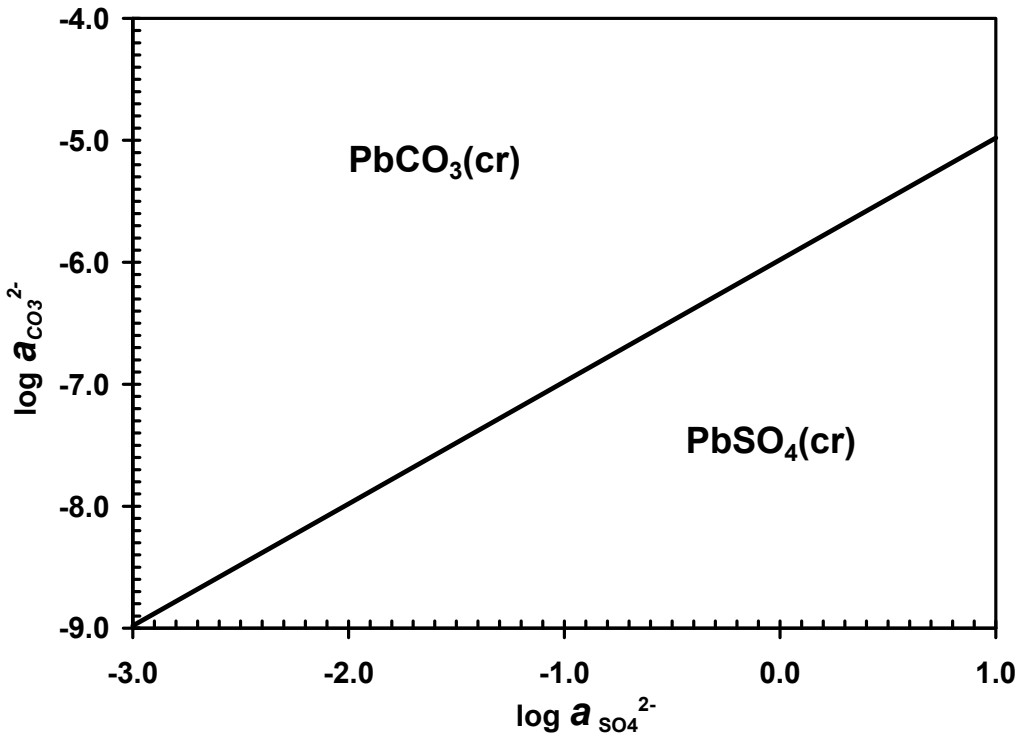

**Figure 12.** The stability fields of lead carbonate [PbCO$_3$(cr)] and lead sulfate [PbSO$_4$(cr)] at 25 °C in the space of $\log a_{SO_4^{2-}}$ versus $\log a_{CO_3^{2-}}$ in carbonate solutions for desulphurization of lead pastes.

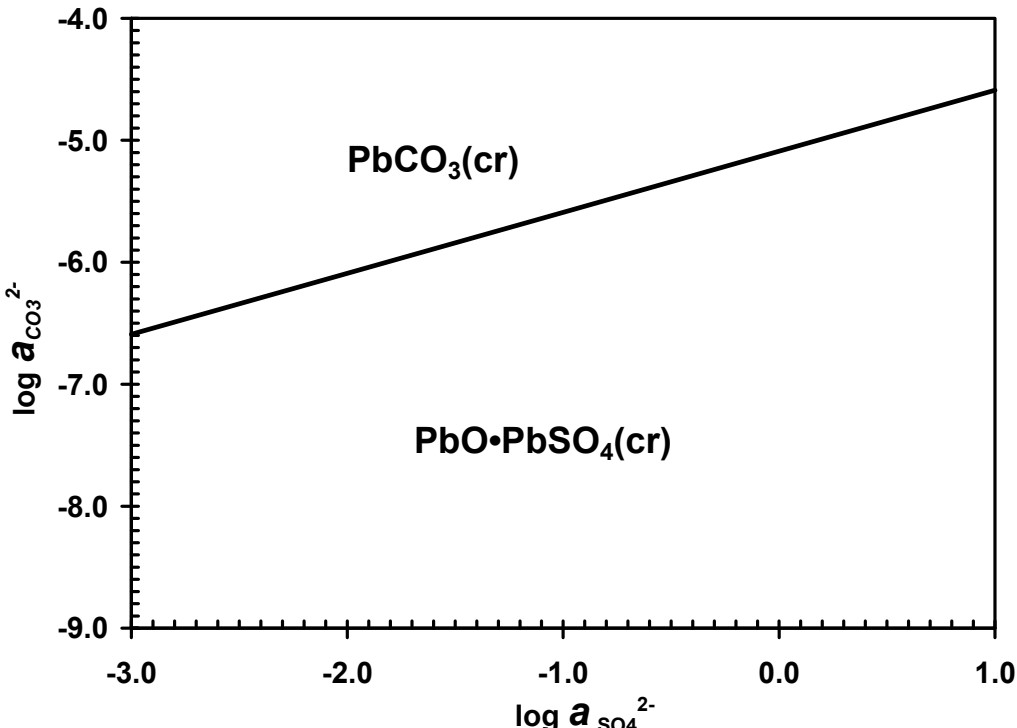

**Figure 13.** The stability fields of lead carbonate [PbCO$_3$(cr)] and lanarkite [PbO•PbSO$_4$(cr)] at 25 °C in the space of $\log a_{SO_4^{2-}}$ versus $\log a_{CO_3^{2-}}$ at pH = 10 and water activity of 1.0 in carbonate solutions for desulphurization of lead pastes.

In carbonate-bearing solutions, lead monoxide present in lead pastes may, or may not, be converted into cerussite. Figure 14 presents the stability fields of cerussite and lead monoxide (PbO, litharge) in the space of activity of the hydroxyl ion in logarithmic units versus activity of carbonate in logarithmic units (i.e., $\log a_{OH^-}$ versus $\log a_{CO_3^{2-}}$). The occurrence or absence of conversion of litharge does not affect the overall desulphurization process. However, from the point of conservation of carbonate solutions, it is advantageous to perform the desulphurization process under the conditions where litharge is stable.

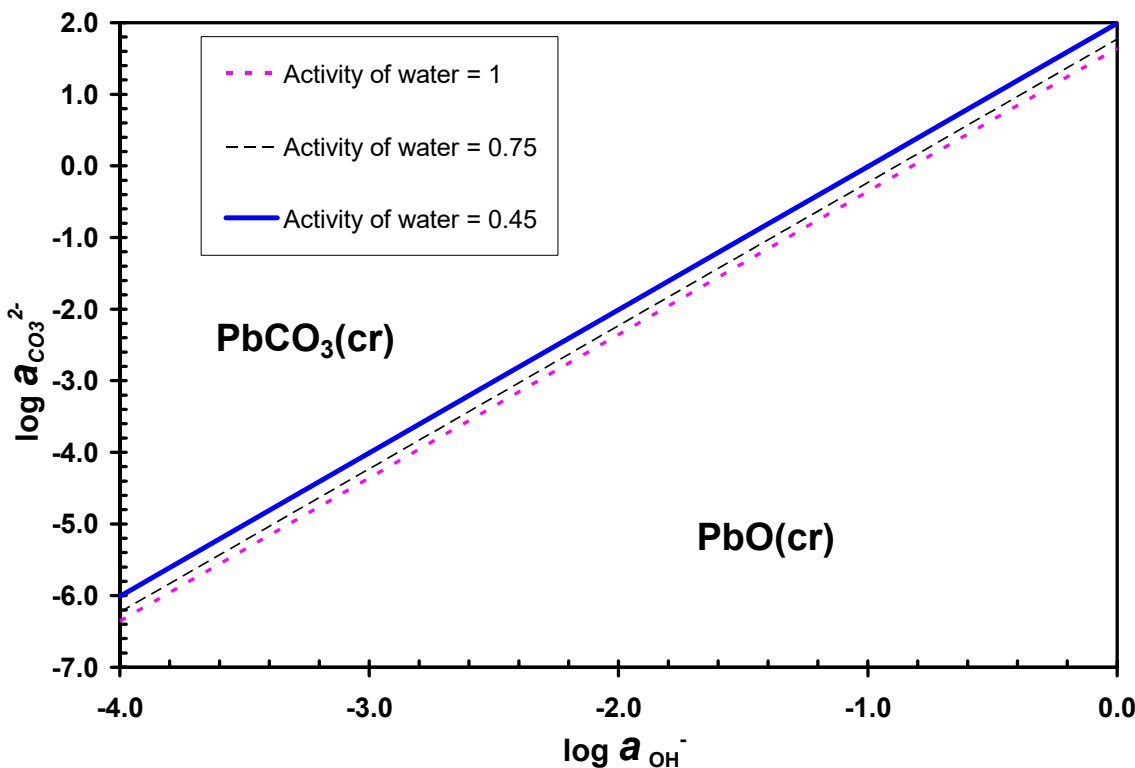

**Figure 14.** The stability fields of lead carbonate [PbCO$_3$(cr)] and lead monoxide [PbO(cr)] at 25 °C in the space of $\log a_{OH^-}$ versus $\log a_{CO_3^{2-}}$ in carbonate solutions for desulphurization of lead pastes.

The carbonate-bearing solutions that can be used for desulphurization of lead sulfates include NaHCO$_3$ alone, mixtures of NaHCO$_3$ and Na$_2$CO$_3$, Na$_2$CO$_3$ alone, and (NH$_4$)$_2$CO$_3$ solutions. The NaHCO$_3$ solutions and mixtures of NaHCO$_3$ and Na$_2$CO$_3$ have been used as supporting solutions for solubility measurements of cerussite before [14], and (NH$_4$)$_2$CO$_3$ solutions have been recently tested for desulphurization [45].

## 4. Discussion

In the desulphurization processes of lead pastes, the transformation or reduction of lead dioxide (plattnerite and scrutiny) is a problem. In the current practice, reductants are needed in reduction of lead dioxide. The reductants that have been used in hydrometallurgical routes include lead sulfide [3] and glucose [35]. Iron powder is also used as a reductant in mechanochemical reduction of lead dioxide [11]. In the mechanochemical reduction process, lead dioxide was reduced under ambient conditions via mechanical ball milling using iron powder as a reductant. On one hand, the use of reductants may increase the cost for the overall desulphurization process. On the other hand, the reduction process via a reductant usually reduces lead dioxide into lead sulfates, which increases the amounts of lead sulfates that need to be desulphurized. In the desulphurization process using oxalate-bearing solutions, there is a strong thermodynamic driving force for the direct conversion of lead dioxide into lead oxalate without an additional reductant, especially in the acidic pH range (see Figure 4). This is related to the stability of lead oxalate. It has been

demonstrated and observed that lead oxalate is a stable phase in the presence of oxalate over a wide range of environments [12,46]. Therefore, oxalic acid solutions are ideal for this direct conversion.

Citrate solutions are also proposed for their use in recycling lead pastes [47–50]. However, the primary function of citrate solutions for recycling lead pastes is that the citrate ion ($C_6H_5O_7{}^{3-}$) forms a strong aqueous complex with lead

$$Pb^{2+} + C_6H_5O_7{}^{3-} \rightleftarrows PbC_6H_5O_7{}^{-} \tag{18}$$

The $\log \beta_1^0$ for $PbC_6H_5O_7{}^{-}$ is 7.28 (see Table 1). The thermodynamic properties for lead citrate solid phases are not well known. Therefore, citrate solutions are currently used for a leaching agent for dissolution of lead solid phases in lead pastes [8]. The interaction parameters of citrate species $PbC_6H_5O_7{}^{-}$, including with major ions such as $Na^+$ and $Mg^{2+}$, are listed in Table 2, which can be used for calculations of solubilities of lead solid phases in citrate solutions.

## 5. Conclusions

The desulphurization of lead pastes is the key process in recycling of lead–acid batteries. In this study, the thermodynamic constraints for three hydrometallurgical routes of desulphurization of lead pastes are presented. The three hydrometallurgical routes of desulphurization include: (1) the conversion of lead sulfates ($PbSO_4$ and $PbO\bullet PbSO_4$) into lead oxalate in oxalate bearing solutions; (2) the conversion of lead sulfates into lead monoxide in alkaline solutions; and (3) the conversion of lead sulfates into lead carbonate in carbonate solutions. Among the above three routes, the desulphurization process via the oxalate route can be performed over the entire pH range, as lead oxalate is stable over the entire pH range from acidic to alkaline regions. As both lead monoxide and lead carbonate are unstable in acidic pH, the desulphurization processes via the alkaline and carbonate routes are preferably performed at activities of the hydroxyl ion higher than $10^{-4.2}$ (i.e., pH $\geq$ 9.8) at 25 °C.

In the oxalate solution route, a minimum oxalate activity of ~$10^{-6}$ is required for the desulphurization. In the alkaline solution route, a minimum hydroxyl activity of ~$10^{-4.2}$ is required for the conversion of lead sulfates into lead monoxide. In the carbonate solution route, a minimum carbonate activity of ~$10^{-6.6}$ is required for the conversion of lead sulfates into lead carbonate.

In light of this study, further investigations would develop effective recycling processes or strategies for lead–acid batteries. For instance, the oxalate route does not require an additional reductant for converting lead dioxide into lead oxalate. Therefore, kinetic studies on the conversion process will provide the important parameters to optimize the conversion of lead dioxide to lead oxalate.

**Funding:** This research received no external funding.

**Institutional Review Board Statement:** Not applicable.

**Informed Consent Statement:** Not applicable.

**Data Availability Statement:** The paper contains all of related data. Therefore, they are already available.

**Acknowledgments:** Sandia National Laboratories is a multi-mission laboratory operated by National Technology and Engineering Solutions of Sandia, LLC., a wholly owned subsidiary of Honeywell International, Inc., for the U.S. Department of Energy's National Nuclear Security Administration under contract DE-NA-0003525. This paper is published with the release number SAND2022-9293J. This paper describes objective technical results and analysis. Any subjective views or opinions that might be expressed in the paper do not necessarily represent the views of the U.S. Department of Energy or the United States Government. The author thanks the editorial office of the *Recycling* journal for the invitation to write this article. The author is grateful to the three journal reviewers for their thorough and insightful reviews, which have significantly helped to improve the manuscript.

**Conflicts of Interest:** The author declares no conflict of interest.

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
