# Peer review of "Recycling of Lead Pastes from Spent Lead–Acid Batteries: Thermodynamic Constraints for Desulphurization"

_recycling, doi:10.3390/recycling7040045_

Round 1

Reviewer 1 Report

The author is to be congratulated on a well presented manuscript. The thermodynamic modelling presented therein provides guidance for future experimental work regarding potential hydrometallurgical treatment routes for lead pastes (waste from end-of-life lead batteries).

The introduction provides a good overview of recent research into potential hydrometallurgical processing routes, and outlines some of the deficiencies of the common industrial practice of pyrometallurgical smelting to recover lead from lead pastes. One point that the author has not addressed (even in general terms) is how the desulfurized lead product (either oxide, carbonate, or preferably oxalate) is to be recycled after desulfurization. Based on the Ellingham diagram for lead, this reviewer suspects that a (presumably simplified) smelting operation to produce lead ingots would be the logical next step in this treatment route, but this is not stated. If the author has other suggestions, they would also fit neatly into this paper.

The thermodynamic modelling has been performed very thoroughly, and the three options (oxalate, oxide and carbonate) have been presented in a good amount of detail in the Results section. The Discussion is concise, and Conclusions suitably supported by the modelled data presented.

There are a few matters this reviewer suggest the author may wish to consider;
1. The Abstract is unusually long (over a page). This could be condensed by, for example, moving the first set of three equations into the Introduction (perhaps around line 88), and paraphrasing all of lines 25 through 33 as "The desulphurization of lead sulfates can be achieved by conversion to lead oxalate, lead oxide, or lead carbonate by appropriate solutions in hydrometallurgical systems." Also, the relevance of the Pitzer model to high ionic strengths is stated twice in the abstract - consider removing the sentence which spans lines 16 to 18.

2. The author alternates between referring to "constraints" and "constrains". In every instance in this manuscript, this reviewer believes the author intended "constraints". Please correct this (it's a simple typo, but the recurring nature was distracting).

3. Table 3, Table 4, and especially Table 5 are long lists of calculated values, which really do not add to the discussion in their current form. It would be convenient to present these as Figures (activities vs reagent concentration) instead of Tables.

4. In line 39, the author claims that no reductant is needed to convert PbO2 to lead oxalate. This reviewer believes the author meant that no additional reductant is needed; in the last equation of Table 1, for example, the H+ ions are the reductant. This same notion is presented in lines 188-190; the author mentions "the direct conversion of lead dioxide into lead oxalate without a reductant", when Figure 4 shows a marked pH dependency for the claimed reaction - this is because the H+ ion is the reductant in this reaction. This reviewer recommends the phrasing be amended to "the direct conversion of lead dioxide into lead oxalate with an added reductant".

5. This reviewer notes with interest the absence of the tetra-hydroxy lead complex [Pb(OH)4]2- amongst the reactions listed in Table 1. Perera, Hefter & Sipos (2001) (An Investigation of the Lead(II)−Hydroxide System | Inorganic Chemistry (acs.org)) report the polarographic and spectrophotometric determination of the cumulative formation constant as -38.0 for this species. The presence of the tetra-hydroxy complex would explain the observed increase in solubility of Pb in high concentrations of NaOH, as shown in Figure 8 (for both the Garrett and Randall data sets cf. the model). Would the author care to comment on why it was seen fit to limit this analysis to weakly alkaline solutions, when lead oxide is well known to be amphoteric? 

Overall, the manuscript is well written, well thought through, and applies known thermodynamic models to the question of battery recycling.

Reviewer 2 Report

This study covers desulphurization behaviors of Pb species based on thermodynamic models with stability constants. The thermodynamic models for solidification of Pb species were considered with various cases. However, these data are based on known information. Also, verification of the proposed models was performed with reported experimental data. Therefore, this study seems to be lack novelty. For instance, when the results of desulfurization under the conditions proposed in this study are shown, the manuscript will be more interesting.

Reviewer 3 Report

1. At title and at various pages including line 88 in page 3 , constrains constraints.  

2. Your abstract should be reduced because it is too long and there are many equations. I think that you may use the expression from your conclusions instead of using several equations.  

3. At line 87 in page 3, you need to add this expression “in the hydrometallurgical recycling process for lead acid batteries, there are three desulphurization processes of lead paste with oxalate, carbonate and alkaline solutions.” 

4. At lines 88-95 in page 3, you need to add this expression “From the optimum condition for each desulphurization route and their products, more efficient and eco-friendly recycling process for lead acid batteries could be examined.” 

5. At the equation(5) of 3.1, you may remove this equation if the reaction(4) is favorable for converting lead dioxide to lead oxalate without reductant.  

6. At your discussion or conclusions, you need to suggest further study based on your results for developing effective recycling process for lead acid batteries.

Round 2

Reviewer 2 Report

This study covers desulphurization behaviors of Pb species based on thermodynamic models with stability constants. The thermodynamic models for solidification of Pb species were considered with various cases. Although this study is first systematic application of thermodynamic models for the desulphurization, overall novelty in this study is limited because the models and data are based on the known information. The authors theoretically found that lead dioxide can be directly converted into lead oxalate without an additional reductant. However, the reduction of lead(IV) with reaction of protons was found elsewhere. Also, whether direct reduction and precipitation of lead with the addition of oxalate or not is not sufficient for the thermodynamic consideration alone. The novelty of this study can be improved with experimental validation.